# Divergent DNA Methylation Signatures of Juvenile Seedlings, Grafts and Adult Apple Trees

**Adrien Perrin [1,*], Nicolas Daccord [1], David Roquis [1,2], Jean-Marc Celton [1,*], Emilie Vergne [1,*] and Etienne Bucher [1,2,*]**

1   IRHS (Institut de Recherche en Horticulture et Semences), UMR 1345, INRA, Agrocampus-Ouest, Université d'Angers, SFR 4207 QuaSaV, F-49071 Beaucouzé, France; nicolas.daccord@gmail.com (N.D.); david.roquis@agroscope.admin.ch (D.R.)
2   Plant Breeding and Genetic Resources, Agroscope, 1260 Nyon, Switzerland
*   Correspondence: adrien.perrin@inra.fr (A.P.); jean-marc.celton@inra.fr (J.-M.C.); emilie.vergne@inra.fr (E.V.); etienne.bucher@agroscope.admin.ch (E.B.)

**Abstract:** The vast majority of previous studies on epigenetics in plants have centered on the study of inheritance of DNA methylation patterns in annual plants. In contrast, perennial plants may have the ability to accumulate changes in DNA methylation patterns over numerous years. However, currently little is known about long-lived perennial and clonally reproducing plants that may have evolved different DNA methylation inheritance mechanisms as compared to annual plants. To study the transmission of DNA methylation patterns in a perennial plant, we used apple (*Malus domestica*) as a model plant. First, we investigated the inheritance of DNA methylation patterns during sexual reproduction in apple by comparing DNA methylation patterns of mature trees to juvenile seedlings resulting from selfing. While we did not observe a drastic genome-wide change in DNA methylation levels, we found clear variations in DNA methylation patterns localized in regions enriched for genes involved in photosynthesis. Using transcriptomics, we also observed that genes involved in this pathway were overexpressed in seedlings. To assess how DNA methylation patterns are transmitted during clonal propagation we then compared global DNA methylation of a newly grafted tree to its mature donor tree. We identified significant, albeit weak DNA methylation changes resulting from grafting. Overall, we found that a majority of DNA methylation patterns from the mature donor tree are transmitted to newly grafted plants, however with detectable specific local differences. Both the epigenomic and transcriptomic data indicate that grafted plants are at an intermediate phase between an adult tree and seedling and inherit part of the epigenomic history of their donor tree.

**Keywords:** epigenetics; perennial plant; transmission of methylation signatures; *Malus domestica*; sexual and asexual propagation

## 1. Introduction

Epigenetic regulation of gene transcription is implemented by several covalent modifications occurring at the histone or DNA level without affecting the DNA sequence itself [1]. These modifications are termed epigenetic marks and can change throughout plant development. Some newly acquired epigenetic changes can also be inherited across generations [2–5]. During their lifetime organisms may develop alternative phenotypes in response to biotic and abiotic stresses [6–9]. These stimuli result in modifications in gene transcription which can be altered by epigenetic modifications [10–12]. Besides gene transcription changes, certain epigenetic marks have been shown to play key roles in DNA conformation and genome stability [2,12,13]. Indeed, DNA methylation has been shown to have

a major role in transposable element (TE) silencing by reducing considerably the potential damage incurred by de novo TE insertions in the genome [14–16].

At the molecular level, DNA methylation consists in the covalent addition of a methyl group to cytosine nucleotide. In plants, DNA methylation occurs in three different cytosine contexts: CG, CHG and CHH (H= A, T or C) [17–20]. DNA methylation is established de novo or maintained by several DNA methyltransferase enzymes [21], each having a specific role depending on the sequence context. In order to maintain DNA methylation following DNA replication that results in hemi-methylated DNA, the methyltransferases MET1 and CMT3 can copy DNA methylation patterns from the "ancestral" strand to the newly synthesized strand. This mechanism is called DNA methylation maintenance [22,23] and occurs at symmetric CG and CHG sequence contexts. However, for the CHH sequence context no such template exists that may allow the DNA methylation maintenance mechanism. In this case, DNA methylation has to be restored by de novo methylation after each DNA replication cycle [24,25]. This pathway is called RNA-directed DNA methylation (RdDM) and requires small interfering RNAs (siRNA) [26,27] to guide the DNA methylation machinery to regions with sequence homology to the siRNAs.

From an epigenetic point of view, perennial plants are of particular interest as they have the potential to accumulate epigenetic modifications throughout their lifetime and may pass this information to the next generation. In addition, in the Rosacea family [28] numerous crops and ornamental plants are multiplied by asexual multiplication via grafting. This is interesting because in addition to the long lifetime of these plants, asexual multiplication involves only mitotic cell divisions [29] and thus presumably increases the chances of transmission of acquired epigenetic marks. If that was the case, epimutations could be quite common in grafted perennial plants. In contrast, during sexual reproduction meiosis can result in epigenetic reprogramming and, therefore, to loss of acquired epigenetic marks [30–32]. In *Arabidopsis*, this reprogramming is the result of active DNA demethylation driven by DEMETER (DME) [30]. Previous studies have suggested that this demethylation process could contribute to the generation of totipotent cells [3,4,33] by alleviating gene silencing via active removal of DNA methylation. These modifications at the DNA methylation level are necessary for normal meiosis [34]. While the RdDM pathway remains active in the egg cell [35], in the central cell of the mature female gametophyte and in matures pollens sperms cells, RdDM activity is reduced [4]. This decrease releases the transcription of TEs, thus resulting in the production of siRNAs derived from those. These siRNAs have been reported to be transported into the egg cell [36] to silence homologous loci in the maternal and paternal genomes [4,36–39]. Based on these findings, one may assume that during sexual multiplication, meiosis would allow restauration of a basal DNA methylation level in these species, while mitosis during asexual multiplication would maintain acquired epimutations.

In plants, inheritance of epigenetic marks has been primarily investigated notably in annual plants. Several studies point to the existence of broad epigenetic variations throughout wild populations of perennial and annual plants [40–42]. Other studies have demonstrated that epigenomic plasticity can allow environmental stress adaptation and improve response to future stresses [43–46]. Furthermore, it has been suggested that epigenetic modifications induced by stress in a mother plant may improve stress response in their offspring [47–49]. However, little is still known about heritable transmission of epigenetic marks in crops and more specifically in woody perennials like apple [29,50,51].

Apple (*Malus domestica*) is a major fruit crop in the world. In 2017, around 130 million tons of fruit were produced on approximately 12.3 million hectares [52]. Because apple is an obligate out crosser, tree propagation for commercial orchards and conservation has to performed via asexual multiplication to maintain the traits of interest. This vegetative multiplication (or clonal multiplication) ensures that all grafted trees originating from a particular cultivar are largely genetically identical. Scions of fruiting cultivars are grafted on rootstock to combine valuable agricultural traits. For instance, in addition to reducing tree size and modifying its architecture, grafting onto particular rootstocks is known to shorten the juvenile phase of the scion by promoting flower differentiation [53]. Scions can thus recover their ability to bloom within 3 to 5 years after grafting [53] while seedlings on their

own roots may only start blooming after up to 8 years [54]. The length of the initial juvenile phase [55] can be highly variable among species, ranging from a few days, as in the *Rosa* genus [56] to more than 30 years in some woody plants [57–59]. Certain phenotypic characteristics have been associated with the juvenile phase such as fast vegetative growth [59], low lignification of young shoots, short internodes, specific leaf shape [55] and low trichome density. For instance, this phenotypic difference between juveniles and adults has previously been described in annual plants such as *Arabidopsis* [60] or *Zea mays* [61], and perennials including the *Acacia* genus, *Eucalyptus globulus*, *Hedera helix*, *Quercus acutissima* [62] or in *Populus trichocarpa* [63].

Here, to better understand how epigenetic marks at the DNA methylation level are transmitted in annual plants, we compared sexual (seed) and asexual (grafting) propagation in apple. Taking advantage of unique homozygous and self-compatible genetic material, we studied how these modes of propagation affect the phenotype, gene transcription and DNA methylation patterns. We present evidence that globally, genome-wide DNA methylation levels are stable in apple independent of the mode of multiplication. However, specific local variations in DNA methylation patterns were detected. These patterns were associated with the regulation of key plant-specific gene regulatory networks such as photosynthesis. This work provides a basis for future studies on the role of epigenetics in tree aging.

## 2. Results

### 2.1. Phenotypic Comparison of Seedlings, Young Grafts and Adult Trees

Unlike most heterozygous apple varieties, the Golden Delicious-derived GDDH13 doubled haploid apple shows a high self-compatibility level. To prevent outcrossing and to produce self-fertilized GDDH13 seeds, we covered trees with insect- and wind-proof cages during blooming time. Then we deployed bumblebees into the cages resulting in the production of hundreds of self-fertilized seeds. This unique material allowed us to study genetically identical seedlings and grafted plantlets derived from the very same parental tree. For that purpose, we simultaneously planted seedlings and grafted budwood from GDDH13 to ensure that the growing plants were of comparable size.

First, we studied the phenotypic differences between parental tree, grafts and seedlings on leaf samples in order to assess whether the plants were in a juvenile or adult phase. Trichome density was the most noticeable phenotypic difference (Figure 1). Leaves sampled from seedlings (SD) displayed a notably lower trichome density on their abaxial face (Figure 1A) compared to the original parental tree (old graft, OG). Leaves sampled from young grafted plants (YG) displayed a similar trichome density as OG (Figure 1B–D).

### 2.2. Transcriptional Profiles of Seedlings, Young Grafts and Adult Trees

In order to identify genes related to the juvenile phenotype or genes displaying differential transcription levels in response to grafting, we performed a set of differential gene transcription analyses. We assessed steady state RNA levels by performing the following two comparisons: OG versus SD (OGvSD) and OG versus YG (OGvYG). Transcriptomes were obtained using a custom-designed microarray that includes probes from all annotated GDDH13 genes and a fraction of transposable elements (TEs). We identified 6943 and 7353 differentially expressed transcripts (DETs) for OGvsSD and OGvsYG, respectively. Of these DETs, 5695 were annotated as genes (differentially transcribed gene, DTGs) in OGvsSD and 4996 in OGvsYG (Figure 2A). In total these DTGs include 13.5% of all annotated genes on the microarray for OGvsSD and 11.8% for OGvsYG (Figure 2A). For transcripts annotated as TEs, we identified 1248 and 2357 differentially transcribed transposable elements (DTTEs) in the OGvsSD and OGvsYG comparisons, respectively (Figure 2B). These represent 5% of all annotated TEs on the microarray for OGvsSD and 6.6% for OGvsYG (Figure 2B).

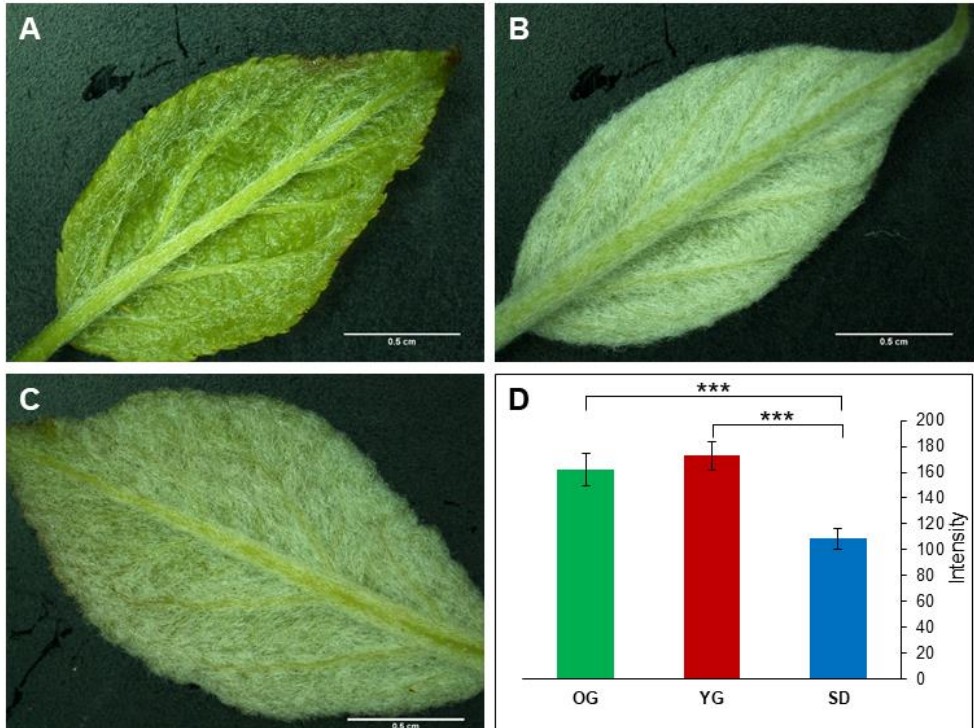

**Figure 1.** Leaf trichome density comparisons between seedlings (SD), young grafted plants (YG) and old graft (OG). Leaf pictures indicate visual differences in trichome density for SD (**A**), YG (**B**) and donor OG (**C**). The graph in (**D**), represents results from light intensity measurements carried out on the abaxial face of leaves. High light intensity correlates with high trichome density. $N = 60$ (5 measurements on 12 leaves) per sample. Statistical differences were evaluated by a two-by-two Kruskall–Wallis test. Asterix *p*-value: *** 1‰.

Overall, DTGs displayed a tendency to be up-regulated in SD and YG compared to OG (Figure 2A). However, for TEs only the OGvsYG comparison followed the same pattern, since up- and down-regulated TEs were more equally distributed in the common DTTEs group. DTTEs specific to OGvsSD displayed a tendency to be down-regulated in SD.

Focusing on the common DTGs between OGvsSD and OGvsYG, we observed two groups (Figure 2A,C). The first group is composed of the 2.085 DTGs displaying a similar regulation pattern: 1365 and 720 DTGs were down- and up-regulated in OGvsSD and OGvsYG, respectively. In the second smaller group, only 85 DTGs displayed an opposite trend: these transcripts were up-regulated in SD in OGvsSD, but down-regulated in YG in OGvsYG. Similarly, we observed two groups for DTTEs (Figure 2B,D). 277 DTTEs were down-regulated in SD and YG in both OGvsSD and OGvsYG respectively, and 226 DTTEs were up-regulated in SD and YG in both comparisons compared to OG. Only 17 DTTEs displayed opposite transcript accumulation patterns compared to the general trend.

To study the main gene regulatory pathways represented in the differential transcription data we used the GDDH13 gene annotation of *Malus domestica* (v1.1) combined with the MapMan software ([64], Figure 3A). We also considered the TE class repartition as previously described in Daccord et al., [65] (Figure 3B). We observed variations in class sizes between OGvsSD and OGvsYG. The most notable variations in size were observed for: photosynthesis (9% of variation in total DTGs in OGvsSD and only 1% in OGvsYG), cell cycle (2% in OGvsSD and 9% in OGvsYG), solute transport (9% in OGvsSD and 4% in OGvsYG), cytoskeleton (1% in OGvsSD and 6% in OGvsYG), RNA biosynthesis (13% in OGvsSD and 18% in OGvsYG), RNA processing (2% in OGvsSD and 6% in OGvsYG) and chromatin organization (2% in OGvsSD and 5% in OGvsYG).

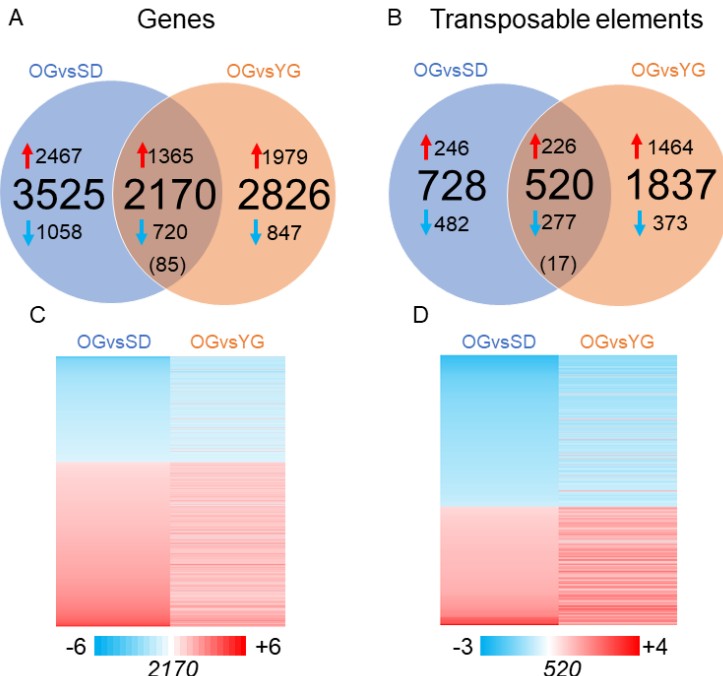

**Figure 2.** Transcriptome comparisons between SD, YG and OG. Graphical representation of the number of differentially expressed transcripts in the different comparisons. (**A**) Venn diagram showing differentially transcribed genes (DTGs) in the comparisons OGvsSD and OGvsYG. (**B**) Venn diagram depicting differentially transcribed transposable elements (DTTEs) in the comparisons OGvsSD and OGvsYG. The central number in brackets represent common transcripts displaying opposed patterns of transcriptional regulation. In (**C**) and (**D**) the heat maps depict transcription ratios of common DTGs (**C**) and DTTEs (**D**). Blue arrow depicting transcripts down regulated in SD and YG compared to OG and red arrow depicting transcript up regulated in SD and YG compared to OG. Numbers of DETs in each heat map are indicated below it. Fold change ratios are shown in the color scale bar.

In order to identify overrepresented classes of genes that could be linked to either the adult or the juvenile phase, we performed an enrichment analysis with the MapMan software using our DTGs as input data (Table S6). In the OGvsSD comparison, seven functional categories were overrepresented including coenzyme metabolism, terpenoids metabolism, chromatin organization, squamosa binding protein (SBP) family transcription factor, protein biosynthesis, peptide tagging in protein degradation and enzyme classification. Eleven classes are overrepresented in the OGvsYG comparison (Table S6), including secondary metabolism, chromatin organization, cell cycle, RNA processing, protein biosynthesis, peptide tagging, cytoskeleton, cell wall, solute transport, and enzyme classification.

Next, we considered the TE class repartition in our DTTE list (Figure 3B). We did not find large variations in the class repartition among comparisons. Class I TE represented 53% of DTTEs on the microarray in OGvsSD and 46% in OGvsYG. Concerning class II TEs we found 31% and 43% of DTTEs in OGvsSD and in OGvsYG, respectively.

Altogether, our analyses show that the sexual and asexual tree propagation methods investigated here had a significant effect on gene and TE transcription in GDDH13.

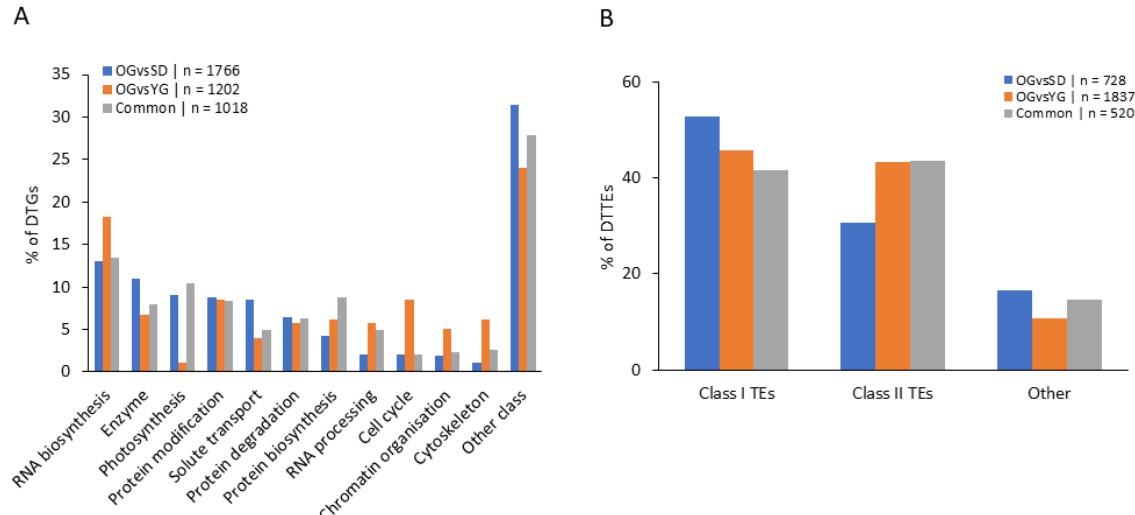

**Figure 3.** Classification of differentially expressed transcripts. (**A**) Percentage of DTGs in each comparison in function of the gene classification according to Lohse et al. [64]. (**B**) Percentage of DTTEs in each comparison in function of TEs classification according to Daccord et al. [65]. Classes represented by less than 5% in the three conditions were summed up in "Other class". Transposon in "other" class correspond to potential host gene and unclassified TEs according to Daccord et al. [65].

### 2.3. Global DNA Methylation Analysis of Seedlings, Young Grafts and Adult Trees

To investigate how DNA methylation marks are transmitted through mitosis as compared to meiosis, we assessed DNA methylation levels in SD, YG and OG samples at the genome-wide level by using whole genome bisulfite sequencing (WGBS). First, we compared genome-wide DNA methylation levels at cytosines in the three sequence contexts (CG, CHG, CHH). Our primary investigation indicated that there was no significant difference in cytosine methylation averages, in any of the contexts among the tested samples (Figure 4A).

Next, we computed and identified differentially methylated regions (DMR) between SD, YG and OG. Overall, we identified 99,437 DMRs in OGvsSD and 44,001 in OGvsYG (Figure 4B). We also investigated DMRs close to genes (Gene-DMRs) or TEs (TE-DMRs). These DMRs are defined by their relative proximity to genes or TEs. For this purpose, we selected DMRs located within 2000 bp 5′ or 3′ of annotated genes or TEs as was previously done in Daccord et al. [65]. We identified 30,902 and 12,150 Gene-DMRs in OGvsSD and OGvsYG, respectively. For TEs, we identified 42,143 and 21,909 TE-DMRs in OGvsSD and OGvsYG, respectively (Figure 4B).

We found that in each comparison, in genes, TEs or other genomic loci, DMRs were largely hypomethylated in SD and YG compared to OG (Figure 4B). Indeed 94% and 63% of DMRs in the three contexts were hypomethylated in SD and in YG respectively compared to OG. Moreover, a vast majority of DMRs were identified in the CHH context (99% and 97% in OGvsSD and OGvsYG, respectively; Table 1). Overall, DMRs in the CHH context tended to be hypomethylated in SD and YG compared to OG (95% in OGvsSD and 66% in OGvsYG) and hypermethylated in SD and YG compared to OG in the CG and CHG contexts (85% in OGvsSD and 93% in OGvsYG) (Table 1). To identify whether DMRs were equally distributed along the genome, or were regrouped within hot spots, we computed the DMR density for the individual contexts as shown in Figure 4C. Overall, we found DMRs to be equally distributed all along the apple chromosomes, with some regions displaying a higher enrichment (Figure 4E, red boxes).

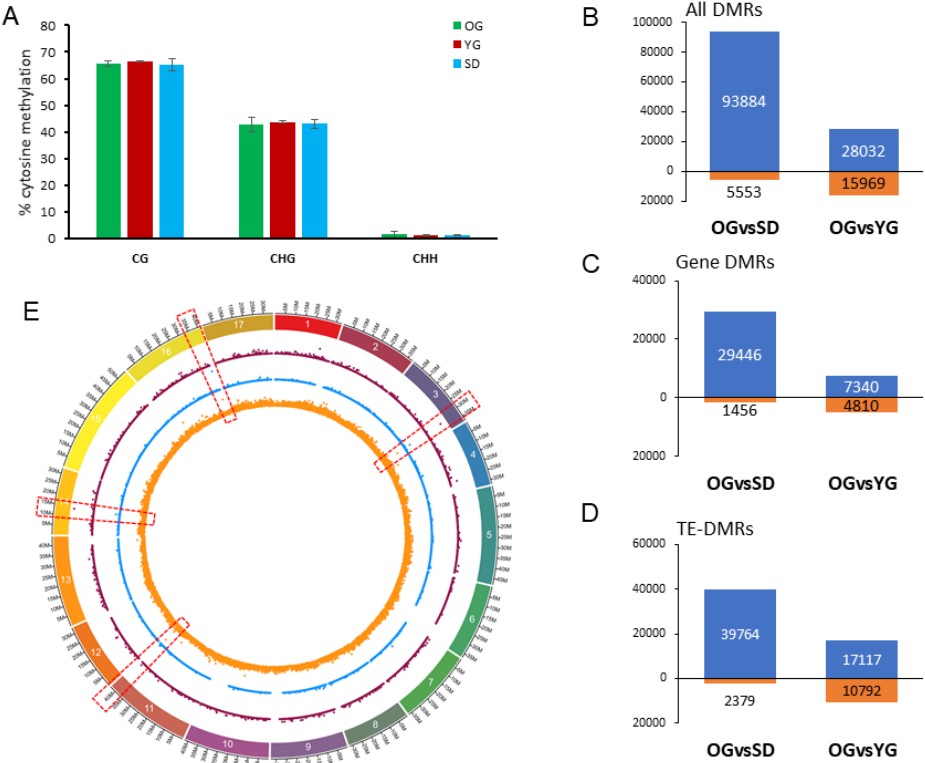

**Figure 4.** Global overview of DNA methylation differences between SD, YG and OG. (**A**) Clustered column chart presenting the genome wide cytosine methylation level (in percentage) of the three methylation contexts (CG, CHG and CHH). Student test was performed to evaluate differences but none was found to be significant. (**B–D**) stacked graph representing the number of differentially methylated regions (DMRs) for each comparison: hypomethylated (above 0, in blue) or hypermethylated (below 0, in orange) in the SD and YG samples compared to OG for all DMRs (**B**), Gene-DMRs (**C**) and TE-DMRs (**D**). DMRs in all sequence contexts were counted and values are indicated in graph. (**E**) Density plot of number of DMRs in 50 kb windows on the GDDH13 genome for OGvsSD (see Figure S1 for OGvsYG). In violet, DMRs in the CG context, in blue for the CHG context and in orange the CHH context. Each point represents the number of DMRs in a 50kb window of the genome. Red dashed boxes indicate the presence of DMR hot spots.

**Table 1.** DMR distributions according to context and methylation changes. Number and percentage of hypo- and hypermethylated DMRs in SD and YG sample compared to OG in each comparison (OGvsSD and OGvsYG). Numbers indicated in grey correspond to the percentage of DMRs in the specific cytosine context in the comparisons. Numbers given in blue indicate the percentage of all DMRs in the corresponding comparison.

| Context | Old Graft vs. Seedling | | | Old Graft vs. Young Graft | | |
|---|---|---|---|---|---|---|
| | Hypomethylated | Hypermethylated | Σ | Hypomethylated | Hypermethylated | Σ |
| | Number (%) | Number (%) | Number (%) | Number (%) | Number (%) | Number (%) |
| CHH | 93.784 (94,9) | 4998 (5,1) | 98.782 (99,3) | 27.945 (65,5) | 14.749 (34,5) | 42.694 (97,0) |
| CHG | 22 (12,3) | 157 (87,7) | 179 (0,2) | 19 (4,9) | 370 (95,1) | 389 (0,9) |
| CG | 78 (16,4) | 398 (83,6) | 476 (0,5) | 68 (7,4) | 850 (92,6) | 918 (2,1) |
| Σ | 93.884 (94,4) | 5553 (5,6) | 99.437 | 28.032 (63,7) | 15.969 (36,3) | 44.001 |

In order to quantify and compare DNA methylation levels we quantified DNA methylation changes (δmC) within DMRs in each sequence context (Figure 5). Overall, we identified significant differences in δmC for the CHG and CHH and not for the CG sequence contexts. Interestingly, in the CHG context, the δmC value was higher in OGvsYG (12.1%) than in OGvsSD (3.6%) for hypomethylated DMRs in YG and SD compared to OG. In the CHH context, we observed that the δmC value was higher in OGvsSD (7.0%) than in OGvsYG (6.1%) for hypomethylated DMRs in YG and SD compared to OG, and lower in OGvsSD (3.7%) than in OGvsYG (7.5%) for hypermethylated DMRs in YG and SD compared to OG. From these results, we conclude that the transmission of cytosine methylation from OG to SD is different to that from OG to YG depending on the cytosine sequence context.

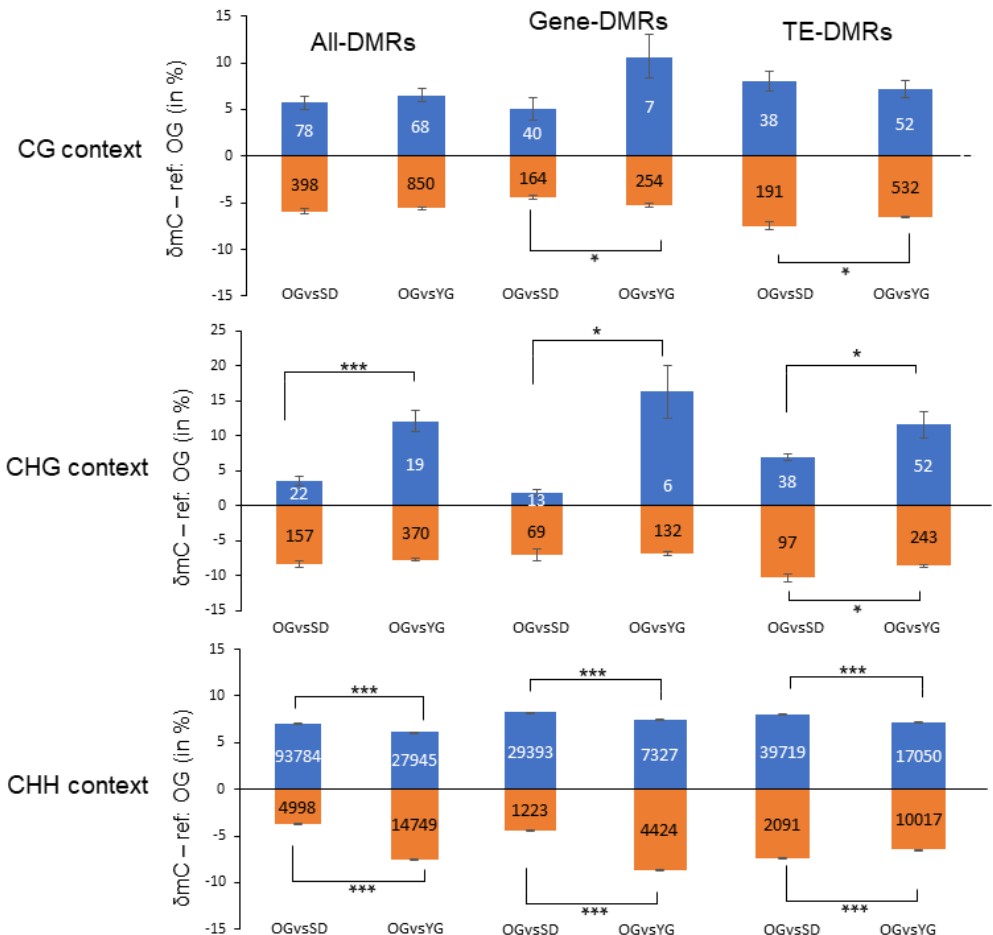

**Figure 5.** Levels of DNA methylation changes in gene and TE annotations. Stacked graph depicting DMR methylation variations (δmC) between samples separated by sequence context and functional annotation. All DMRs taken together are presented in the All-DMRs column, DMRs at genes and TEs in the Gene-DMRs and TE-DMRs columns, respectively. DMRs were filtered by *p*-value and SDA (standard deviation average) in accordance to a fixed threshold (Table S2). Student test was performed to evaluate differences in δmC, results are represented by an asterisk depending on the *p*-value threshold: * 5%; ** 1%; *** 1‰. δmC: delta of methylation. In blue and orange, the hypomethylated and hypermethylated DMRs in SD and YG compared to OG, respectively.

For DMRs located in genic regions (Gene-DMRs, Figure 5) we observed that there were less DMRs in the CG-CHG (286 for OGvsSD and 393 for OGvsYG) contexts than in the CHH context (30,616 for OGvsSD and 11,751 for OGvsYG). Gene-DMRs in the CG and CHG contexts were almost all hypermethylated in YG and SD compared to OG in both comparisons. Indeed, 80% and 99% of Gene-DMRs in the CG context were hypermethylated in OGvsSD and OGvsYG, respectively. 84% and

96% of Gene-DMRs were hypermethylated in CHG in OGvsSD and OGvsYG, respectively. This is consistent with the observations we made for the All-DMRs group (Table 1). Conversely, 96% and 62% of Gene-DMRs in the CHH context were hypomethylated in SD and YG compared to OG for OGvsSD and OGvsYG, respectively.

While studying the DNA methylation changes, we found that in the CG and CHH contexts, the δmC values of hypomethylated Gene-DMRs were smaller in OGvsSD (4.2% for both comparisons) than in OGvsYG (4.9% and 8.2%, respectively). However, for hypomethylated Gene-DMRs in the CHG context in YG or SD compared to OG, the δmC value was higher in OGvsYG (16.3%) than in OGvsSD (1.8%) following the overall trend observed for All-DMRs. Then, for hypomethylated Gene-DMRs in the CHH context in YG or SD compared to OG, the δmC value was higher in OGvsSD (7.7%) than in OGvsYG (6.9%) following the trend observed for All-DMRs. These observations indicate towards a contrasted sequence context-specific pattern of DNA methylation differences.

For DMRs located in TE annotations (TE-DMRs, Figure 5), our observations were similar to the results for Gene-DMRs. Overall, most TE-DMRs were hypermethylated in YG or SD compared to OG in the CG (83% for OGvsSD and 91% for OGvsYG) and CHG (72% for OGvsSD and 82.4% for OGvsYG) contexts, and hypomethylated in the CHH context (95% for OGvsSD and 63% for OGvsYG). As for Gene-DMRs we found only differences to hypermethylated TE-DMRs in CG context, but here, δmC value is higher in OGvsSD (7%) than in OGvsYG (6.1%). For CHG, the δmC value of hypomethylated TE-DMRs was smaller in OGvsSD (6.9%) than in OGvsYG (11.6%) and higher for hypermethylated TE-DMRs in OGvsSD (10.3%) as opposed to OGvsYG (8.6%). Finally, in CHH context, we observed that δmC value was higher in hypomethylated and hypermethylated TE-DMRs in OGvsSD (7.5% and 7%, respectively) than in OGvsYG (6.7% and 6.1%, respectively).

Overall, even though there were no large global differences in DNA methylation level between the samples analyzed here, we found significant local differences. The majority of DMRs were in the CHH context with a tendency to be hypomethylated in YG and SD compared to OG.

### 2.4. Classes of Genes Enriched with Differentially Methylated Regions (DMRs)

To identify genes belonging to particular functional categories and presenting DMRs in their proximity, we used the aforementioned GDDH13 annotation in MapMan and the TE annotation as previously used in our transcriptomic analysis. Here, we only considered Gene-DMRs and TE-DMRs in the CHH context. We excluded DMRs associated with the CG and CHG context from further analyses due of their very limited number (Tables S2 and S3). For the following, we termed as DTG-DMRs genes that we found to be differentially transcribed and containing or being close to DMRs. Similarly, TEs identified as DTTEs and being associated with TE-DMRs were termed DTTE-DMRs.

As expected, we found the seven classes that we previously identified in DTGs analysis: RNA biosynthesis, protein modification, enzyme family, protein degradation, solute transport, photosynthesis and protein biosynthesis (Figure 6A). We did not find differences in the proportion of gene classes between OGvsSD and OGvsYG.

For DTTE-DMRs (Figure 6C) we observed a smaller proportion of Class I TEs in OGvsYG (57.4%) compared to OGvsSD (77.7%), while for class II TEs we found 35.4% for OGvsYG and 15.5% for OGvsSD.

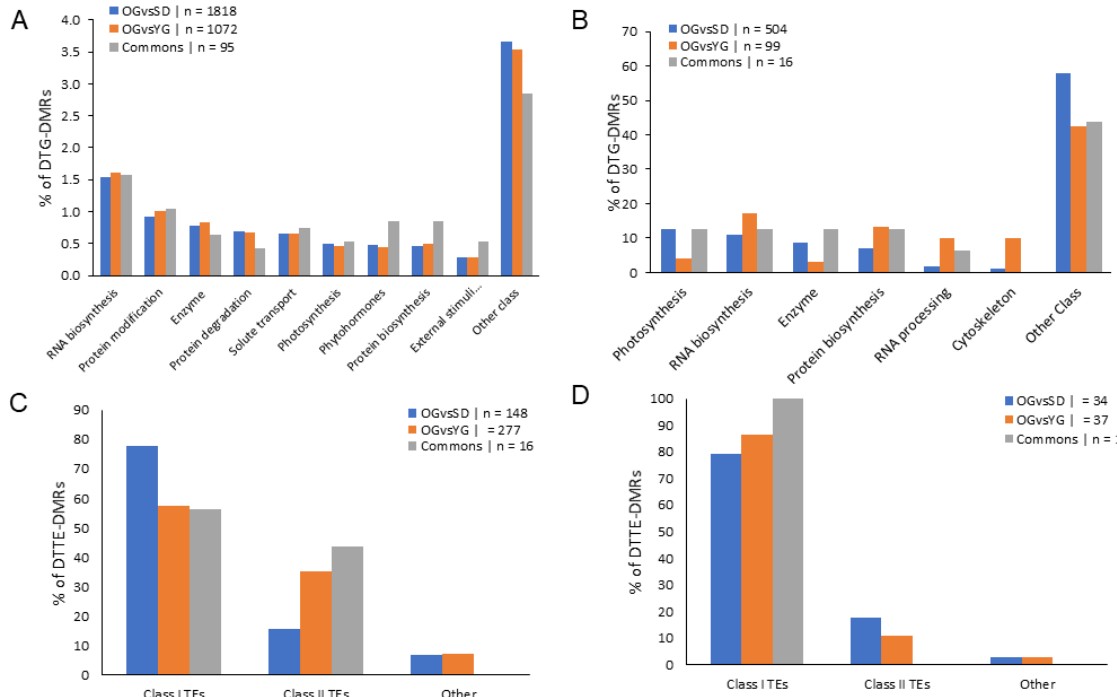

**Figure 6.** Classification of differentially transcribed genes that are associated to DMRs. Clustered column charts depicting the percentage of DTG-DMRs (**A** and **B**) and DTTE-DMRs (**C** and **D**) in the respective comparisons in function of gene or TEs classification. Only DMRs in the CHH context are presented here. In (**A**) and (**C**) all DTGs- and DTTEs-DMRs were used while in (**B**) and (**D**) we only considered DTGs and DTTEs with differential transcription ratio greater than 1.5 in absolute value. Gene classes representing less than 5% (**A**) or 10% (**B**) of the total in the three conditions were summed up in "other class".

## 2.5. Relationship between DNA Methylation and Transcription

In order to identify correlations between DNA methylation and gene expression we associated Gene- and TE-DMRs to our microarray transcriptome data. The gene classes were defined according to the Mapman annotation of genes and to the TE annotation previously used to analyze DTGs and DTTEs.

We found 520 DTG-DMRs in OGvsSD and 115 DTG-DMRs in OGvsYG (Figure 6C), 35 DTTE-DMRs in OGvsSD and 38 DTTE-DMRs in OGvsYG (Figure 6D). We investigated genes and TEs classes' repartition for DTG- and DTTE-DMRs. Of the 11 classes found in DTGs analysis (Figure 3A), we found six gene classes representing only slightly more than 5% of all DTG-DMRs. These included the classes photosynthesis, RNA biosynthesis, enzyme family, protein biosynthesis, RNA processing and cytoskeleton (Figure 6B).

We did not observe notable shifts within the classes' repartition between OGvsSD and OGvsYG for DTTE-DMRs (Figure 6D).

Finally, we investigated the link between DMRs and DTGs. Again, we only considered DTG-DMR in the CHH context due to the low number of DTG-DMR we found in the CG and CHG contexts. We noticed that in both comparisons the majority of DMRs in DTG-DMRs were located in gene promoters (553 for OGvsSD and 126 for OGvsYG), followed by the terminator region (284 for OGvsSD and 71 for OGvsYG) finally followed by those present in gene bodies (111 for OGvsSD and 26 for OGvsYG) (Figure 7A,B. See examples of these DTG-DMRs in Figure S2).

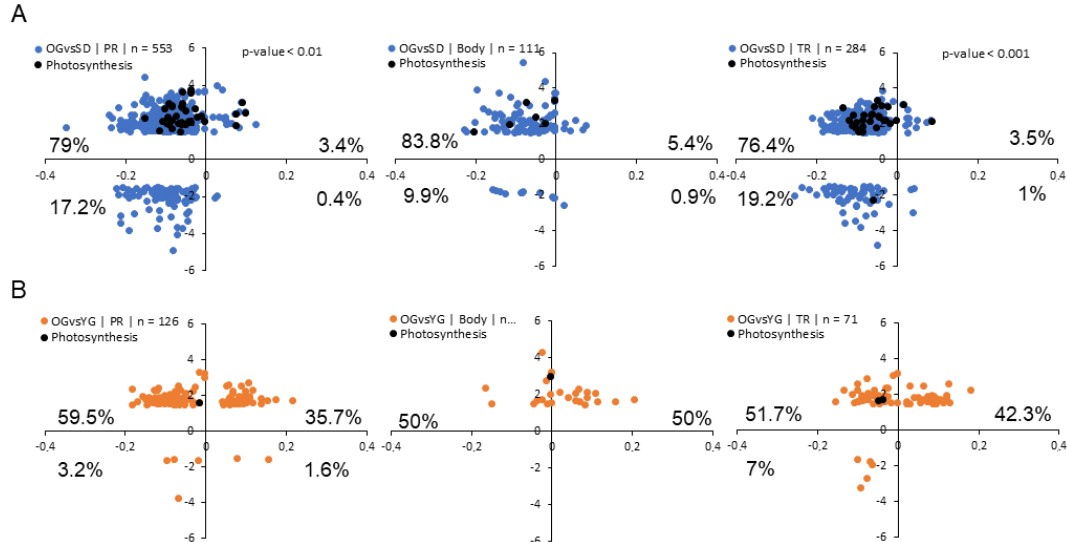

**Figure 7.** Relationship between transcription ratio and DNA methylation variation Scatterplot representing DTG-DMRs in OGvsSD (**A**) and OGvsYG (**B**) in the CHH context, X axis represents δmC and the Y axis represents gene expression ratios of the corresponding comparisons. In blue/orange are shown all DTG-DMRs and in black the ones specifically associated to photosynthesis. Numbers of DTG-DMRs used in each graph are indicated in the legend and corresponding percentages in each quadrant at the edges of the graph. We separated DTG-DMRs in function of the position of the DMRs related to the corresponding gene (PR = promoter, body, TR = terminator). Student test was performed to evaluate the correlation between expression and DMR state, significant correlations are indicated with *p*-values on corresponding scatterplot.

Our data also indicates that, hypomethylated DMRs located in promoter and terminator regions of genes, were associated with transcriptional up-regulation in SD compared to OG (Figure 7A) but we did not find that correlation in the OGvsYG comparison.

Among the classes of DTGs associated with DMRs, we observed that genes associated with photosynthesis tended to be both hypomethylated and up-regulated in SD or YG compared to OG. Indeed 92% of Gene-DMRs associated with the photosynthesis pathway presented this pattern in OGvsSD and 100% of them in OGvsYG.

Our results indicate that in the CHH context, hypermethylation of a DNA sequence in the proximity of a gene correlates with a reduced level of transcription of that particular gene following sexual propagation.

## 3. Discussion

### 3.1. Newly Grafted Plants Are at an Intermediate State between Adult Tree and Juvenile Seedling

Phenotypic differences between juvenile and adult plants are commonly observed at the leaf level [55]. In our study we observed that leaves of apple seedlings displayed a low trichome density compared to the donor tree (Figure 1). In contrast, young grafted plants did not display a difference in trichome density compared to the donor tree. As previously reported by others, this phenotype can be associated to the juvenile phase [66] and a young grafted plant is thus developmentally closer to the adult tree than to the juvenile seedling. Nonetheless, newly grafted plants show a contrasted ability to flower. In the grafting process, a mature bud (able to flower or quiescent) is placed on a short-rooted stem (rootstock). The number of nodes between the apical bud and the rootstock is drastically reduced to 1 or 2 nodes. After their first year of growth, buds are in a mature adult state but are unable to initiate flowers and to bear fruits because of an insufficient number of nodes (less than 77) in the stem.

This limit was previously described as a transition phase between juvenile and adult apple tree [67–69]. Thus, young grafted plants are not adult plants from a physiological point of view.

Here, we wanted to study the molecular changes that occur during propagation via grafting and by seed formation. First, we compared the transcription profiles in three different stages: seedlings (SD), young grafted plants (YG) and an adult tree (old graft, OG), taking advantage of our genetically identical material growing under controlled conditions. Globally we observed a higher transcription level for the majority of the DTGs and DTTEs in the SD or YG compared to OG. This correlates well with the previously reported decrease in gene transcription in mature compared to juvenile plants [70–72]. Furthermore, the common DTGs identified as activated in the OGvsSD and OGvsYG comparisons could be correlated to high vegetative growth in younger stages. Such rapid growth can be observed in seedlings and in newly grafted plants, and in mature apple trees gene expression is generally reduced, as observed in Day et al. [73]. Thus, the transcriptome of newly grafted plants showed similarities with the one obtained from seedlings but also with that of adult trees. These observations are in line with previous studies on other woody plants [70–73].

Overall, our findings indicate that young grafted plants are at the interface between a juvenile seedling and an adult mature tree [67,68] from a morphological and transcriptomic perspective.

This intermediate condition of newly grafted plants is also confirmed from a molecular point of view. Indeed, we identified differences in gene class repartition for DTGs between OGvsSD and OGvsYG (Figure 3A) which included photosynthesis, RNA processing, chromatin organization and cell cycle classes. As regards genes related to photosynthesis, we found that they represented 9% of the DTGs in OGvsSD but only 1% in OGvsYG. This is consistent with previous observations showing that the photosynthetic pathway is differentially regulated between juvenile and mature reproductive plant, especially in woody plants (reviewed in [74]). Indeed, photosynthesis is known as a physiological process subjected to many modifications during the transition from the juvenile to the mature phase [75]. As juveniles, SD undergo broader transcriptomic changes compared to YG and OG. This can be associated to an age-related gene transcription pattern previously described for photosynthesis-related genes in other woody plant such as *Pinus taeda* [76], *Larix laricina* [77], *Picea rubens* [78] and in the *Quercus* genus [79].

This intermediate condition of newly grafted plants between juvenile seedlings and adult tree was also observed at specific loci at the DNA methylation level in the CHH context. Indeed, overall a hypomethylation of the CHH-DMRs was observed in SD andYG compared to OG. We also found a less extensive hypomethylation in OGvsYG (65.5% of all DMRs were hypomethylated in YG compared to OG) than in OGvsSD (95% of all DMRs were hypomethylated in SD compared to OG).

## 3.2. DMRs Correlate with Neighboring Gene Transcription

Previous reports established a correlation between DNA methylation and the repression of gene transcription, particularly in *Arabidopsis* [80,81]. In this study, we investigated a possible link between DNA methylation and gene transcription changes in *M. domestica*. For that purpose, we associated DMRs with their neighboring DTG in order to investigate the effect of methylation on gene transcription. We found that in the CHH context, genes with closely located hypomethylated DMRs in promoter or terminator regions in SDs compared to OG often also displayed a higher transcription level (Figure 7). This was particularly the case for photosynthesis-related genes (Figure 7). Our correlative observations thus indicate that cytosine methylation in the CHH context may to be involved in regulating the transcription of these genes.

We did not only observe local changes in DNA methylation, but also contrasted levels of DNA methylation changes ($\delta$mC). Indeed, we found significant differences at the $\delta$mC level between both comparisons, particularly in the CHG and CHH contexts (Figure 5). For methylation in the CHH context, we observed that, even if the difference in $\delta$mC was significant between OGvsSD and OGvsYG, it was not very high between the comparisons but also within comparisons. Indeed, the highest $\delta$mC was on average above 8% for hypermethylated Gene-DMRs in OGvsYG. But we also found that

this relatively small methylation variation was sufficient to relate it with gene transcription changes (Figure 7).

## 4. Materials and Methods

### 4.1. Plant Material

*Malus domestica* plants were obtained from GDDH13 [82] line (X9273). Young grafted plants (YG) were obtained by grafting budwood of GDDH13 orchard tree (planted in 2001) on the rootstock cultivar 'MM106'. Seedlings (SD) were obtained by self-fertilizing GDDH13 flowers on an isolated tree located in INRAE's orchards. Seed dormancy was removed after a 3 months period of cold stratification before sowing. The homozygous state of the seedlings used in this study was evaluated using publicly available SSR markers. A clone of the original GDDH13 tree, grafted onto an 'MM106' rootstock in 2007 and placed in the greenhouse in 2016 was used as reference mature adult tree (old graft, OG).

### 4.2. Phenotyping

Nine young leaves were collected for each sample (old graft, seedling and young graft) and time point. At each sampling time SD and YG plants were pruned to increase vigor. Three biological replicates were used at three weeks intervals for YG and SD materials in 2018, and one replicate was collected in 2019 including OG material (twelve leaves were sampled). Each leaf was then photographed using binocular magnifier (Olympus SZ61, Schott KL 1500 LED, Olympus DP20). Pictures were further analyzed with the ImageJ® software [83]. Pictures were transformed in 8-bit grayscale and light intensity was measured on 5 areas of 0.03 cm$^2$ on each leaf. Intensity differences among samples were evaluated using the R language by the Kruskal–Wallis test. We first analyzed biological replicates harvested in 2018 and 2019 (SD and YG). Because there were no differences among biological replicates of SD in 2018 and 2019, and similarly for YG sample in 2018 and 2019, we present only the results of 2019, which also include the OG sample.

### 4.3. DNA and RNA Extraction

The youngest and completely opened leaf below the apex was sampled for each replicate. Sampling was performed as described in Table S1. DNA was extracted using NucleoSpin Plant II kit (Macherey-Nagel, Hoerdt, France). The manufacturer's recommendations were applied with the following modifications: at step 2a PL1 buffer volume was raised to 800 µL and PVP40 was added (3% of final volume), suspension was then incubated 30 min at 65 °C under agitation. The lysate solution was centrifuged 2 min at 11,000 g before transferring the supernatant in step 3. At step 4 PC buffer volume was raised to 900 µL. In step 6 the first wash volume was decreased to 600 µL and the third wash volume was raised to 300 µL. An extra-centrifuge step was added after washing to remove ethanol waste from the column. In step 7 DNA was eluted twice in 55 µL in total. The RNA was extracted using the NucleoSpin® RNA kit (Macherey-Nagel, Hoerdt, France) according to the manufacturer's protocol.

### 4.4. Bisulfite Sequencing and DMRs Calling

Extracted DNA was precipitated using a mix of pure ethanol (70%), water (24%) and NaAc 3M (3%). After precipitation DNA was sent to Beijing Genomics Institute (Shenzhen, Guangdong 518083, China) in pure ethanol for whole genome bisulfite sequencing (WGBS). An average depth coverage of 14X per sample was obtained. DNA methylation data can be accessed on the Gene Expression Omnibus website under accession codes GSE138377. Bisulfite sequencing reads were mapped on GDDH13_V1.1 reference genome with Bsmap tool [84] to obtain methylation calling file. Methylation averages between samples were compared by Student t-test using R [85].

We named differentially methylated regions (DMRs) using a pipeline developed in IRHS (Angers, France) published at GitHub platform (https://github.com/EtienneBucher/DAMOCLES). DMRs were

calculated between OG and SD samples and between OG and YG samples with the following parameters: minimum coverage of 3×, 200 bp sliding windows with 100 bp overlaps. DMRs files contain quality values such as *p*-value, standard deviation of averages (SDA) and methylation differences. SDA evaluates the natural variation between biological replicate, a high variation between replicates gives high SDA value. We empirically determined a threshold for each DNA methylation sequence context using visual inspections of DMR preview in JBrowse [86]. SDA thresholds were fixed as mentioned in Table S2. Thresholds were determined in order to select the most reproducible DMRs within biological replicates (Table S3; GSE138377).

In order to evaluate the variation of methylation levels within DMRs, we extracted DNA methylation level difference of YGvsOG and SDvsOG. Student test was performed to evaluate difference in methylation levels.

### 4.5. Microarray

The *Malus domestica* microarray (Agilent-085275_IRHS_Malus_domestica_v1; GPL25795; Agilent, Foster City, CA, USA) was used for transcriptome analysis. Details about this array are available on Gene Expression Omnibus at following link: https://www.ncbi.nlm.nih.gov/geo/query/acc.cgi?acc=GPL25795. Briefly, this array contains 89,218 coding DNA sequences (sense and antisense), 582 miRNAs (sense and antisense) and 13,834 ncRNAs (sense and antisense). This array also includes all annotated TEs families consisting in 69,966 transposable elements probes (TEs, sense and antisense). Complementary DNA (cDNA) were synthesized and hybridized with the Low Input Quick Amp Labeling Kit, two-color (Agilent, Foster City, CA, USA). Two biological replicates were used. Each biological replicate represents one sample for OG and YG materials, and a pool of two samples for SD material. Hybridizations were performed on a NimbleGen Hybridization System 4 (mix mode B) at 42 °C overnight. Slides were then washed, dried, and scanned at 2 μm resolution. NimbleGen MS 200 v1.2 software was used for microarray scans, and the Agilent Feature Extraction 11.5 software was used to extract pair-data files from the scanned images. We used the dye switch approach for statistical analysis as described in Depuydt et al. [87]. Analyses were performed using the R language (R Development Core Team, 2009); data were normalized with the lowess method, and differential transcription analyses were performed using the lmFit function and the Bayes moderated test using the package LIMMA [88]. Transcriptomic data are available in Gene Expression Omnibus website, with the accession GSE138491.

### 4.6. Reverse Transcriptase Quantitative Polymerase Chain Reaction (RT-qPCR) Microarray Validation

Extracted mRNA was treated by DNAse with the RQ1 RNase-Free DNase (Promega, Madison, WI, US) following the manufacturer's protocol. The Moloney Murine Leukemia Virus Reverse Transcriptase was used to obtain cDNA from 1.2 μg of total RNA, with oligot(dT) primers following the manufacturer's protocol (Promega, Madison, WI, USA). For quantitative polymerase chain reaction (qPCR) measurements, 2.5 μL of cDNA at the appropriate dilution were mixed in a final volume of 10 μL with 5 μL of quantitative PCR mastermix (MasterMix Plus for SYBR Green I with fluorescein; Eurogentec EGT GROUP, Seraing, Belgium), with 0.2 μL of each primer (200 nM final) and with 4.1 μL of pure water. Primers were designed with Primer3Plus [89] and were used at their optimal concentration using the reaction efficiency calculation (near to 100%) according to Pfaffl recommendations [90]. Genes used to validate the microarray data were selected in differentially transcribed gene (DTG) lists in both comparisons (OGvSD and OGvYG) with 1) a high ratio value and 2) high intensities values. Accessions and primer sequences are indicated in Figure S3A. Reactions were performed with a CFX connect real-time system (Bio-Rad, Hercules, CA, USA) using the following program: 95 °C, 5 min; 35 cycles comprising 95 °C for 3 s, 60 °C for 45 s; 65 °C, 5 s and 90 °C for 1 min, with real-time fluorescence monitoring. Melting curves were acquired at end of each run. Data were acquired and analyzed with CFX Maestro V1.1 (Bio- Rad, Hercules, CA, USA). Gene transcription levels were calculated using the $2^{-\Delta\Delta Ct}$ method and were corrected as recommended by Vandesompele et al. [91] (Figure S3B), with three reference genes: *ACTIN* (accession CV151413, MD14G1142600), *GAPDH* (accession CN494000,

MD16G1111100), and *TUBULIN* (accession CO065788, MD03G1004400) used for the calculation of a normalization factor.

*4.7. Differentially Expressed Transcript (DET) Analysis*

Differentially expressed transcripts were selected based on their *p*-value ≤1% (Table S4). For DET other than TEs and miRNA a MapMan annotation (https://mapman.gabipd.org/home; version 3.5.0BETA), was performed, using GDDH13_1-1_mercator4 map file, in order to assign each DET to a class of genes (BIN code). DET not assigned to a BIN class were excluded. A representativeness percentage of each BIN class was then calculated in the comparisons OGvSD, OGvYG and in the intersection between the both comparisons. A MapMan enrichment analysis using the BIN class representativeness was performed and a BH correction was applied [92] because of the high number of values. The DETs associated to a gene annotation are called DTGs in this manuscript. For differentially transcribed TEs (DTTEs), the TE classification [65] was used in order to assign each DTTE to a class.

*4.8. Association between DMR and Differentially Transcribed Gene (DTG) or Differentially Transcribed Transposable Element (DTTE)*

DMRs and transcription level data (DTG or DTTE) results were connected thanks to gene identifier. DMRs without associated DETs were removed. There is some redundancy in the gene or TE identification because several DMRs could be located to the same gene or TE. To avoid biases in our analysis we only kept DMRs with the highest methylation variation for each gene or TE (Table S5). For this analysis we applied a threshold and kept only transcripts with differential expression ratios above 1.5 and below −1.5 in order to better identified pathways or genes to work with. In order to identify correlation between DNA methylation and gene expression, we extracted the coefficient of determination of the linear regression of each scatterplot (Figure 7). We calculated the square root of the coefficient of determination and performed a Student test. Next, we compared this value to the theoretical value of the Student test. If a correlation was found, we indicated *p*-values on the scatterplot. Because of the low number of values ($n < 50$) in gene body localization in OGvYG comparison, we first performed a Fisher test that validated the homogeneity of variance, thus allowing us to perform a Student test on these values.

## 5. Conclusions

In this study we compared the transmission of epigenetic marks and their potential effects on transcription during sexual and asexual propagation in apple.

First, we identified a phenotypic change (Figure 8) that was associated with adult plant phase and confirmed that grafting is not comparable to a complete rejuvenation process, as observed in seedlings. In our transcriptomic analysis we notably found that the transcription level of genes related to photosynthesis was elevated in seedlings compared to the tree, while newly grafted plants displayed an intermediate transcription level (Figure 8).

Analysis of the methylation data indicated that at the genome scale, the level of methylation in all three samples was similar. However, we were able to identify DMRs particularly in the CHH context. This result indicates that DNA methylation reprogramming during meiosis may not affect the global methylation level of the genome, but rather modifies particular regions of the genome. This observation was particularly striking regarding genes associated with photosynthesis. As found in the transcriptomic analysis, the DNA methylation data also indicated that grafted plants were at the interphase between the tree and the seedlings.

Globally, our results indicate that, from a physiological, transcriptomic and epigenomic standpoint, newly grafted plants are at the interphase between a tree and a seedling, displaying characteristics that are particular to both the mature and the young immature stages of the plant.

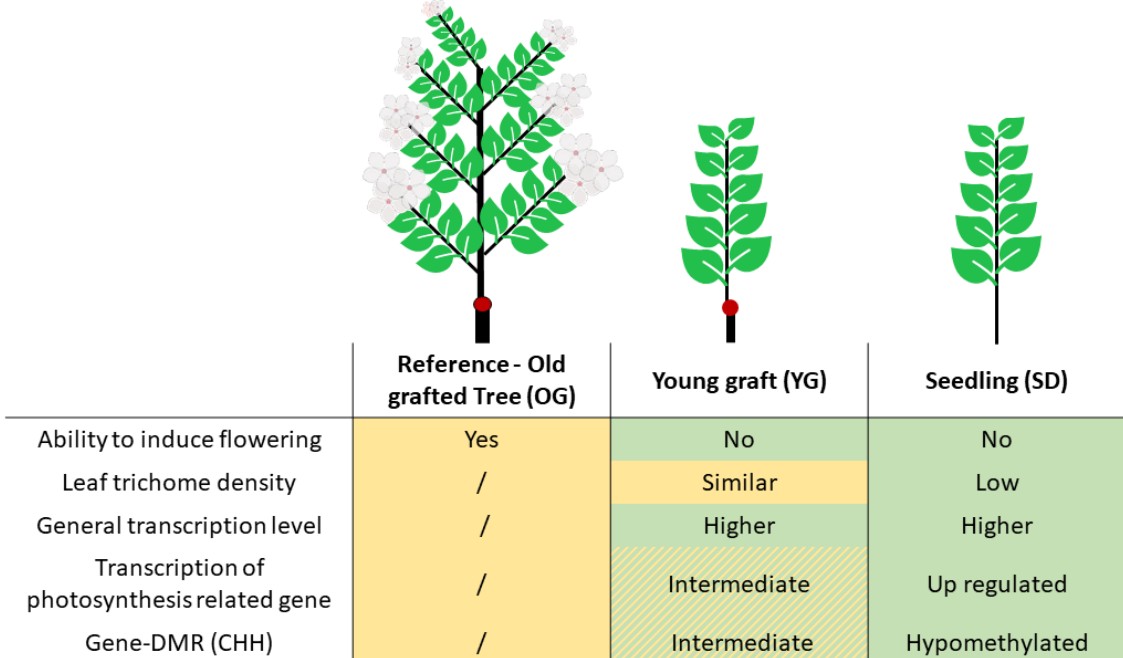

**Figure 8.** General overview of the main results of this study concerning physiological and molecular changes occurring during sexual and an asexual propagation. The red dot represents the grafting point between scion and rootstock (larger line weight). Shared aspects between plants are highlighted by the background colours.

**Supplementary Materials:** The following are available online at http://www.mdpi.com/2075-4655/4/1/4/s1: Figure S1: Microarray validation, Figure S2: Methylation overview in GDDH13, Figure S3: Jbrowse screenshoot, Table S1: Resume of defined samples and details of sampling, Table S2: Fixed threshold to filter DMRs calculated between each comparison, Table S3: DMRs list of OGvsSD and OGvsYG comparisons, including Gene- and TE-DMRs, Table S4: DETs list of OGvsSD and OGvsYG comparison, Table S5: DTG- and DTTE-DMRs list of OGvsSD and OGvsYG comparisons, Table S6: Functional classes found in enrichment analysis on Mapman software, Table S7: Count of DTG-DMRs in OGvsSD and OGvsYG comparisons, Table S8: Count of DTTE-DMRs in CHH context in OGvsSD and OGvsYG comparisons. Data Archiving Statement (secure tokens were provided to editor to allow reviewers' access). WGBS: methylation calling files, DMRs list and transcriptomic data were deposited on Gene Expression Omnibus online platform under the accession number GSE138492. We obtained two sub-accession numbers. The first, GSE138377, comprising all data related to methylation analysis as WGBS data, processed data and an excel spreadsheet of calculated and filtered DMRs to OGvsSD and OGvsYG comparison named Supp_Tab_S3.xlsx. The second, GSE138491, comprising the transcriptomic data obtained by microarray as describe in method.

**Author Contributions:** Conceptualization, J.-M.C. and E.B.; Formal analysis, A.P., N.D., D.R., J.-M.C., E.V. and E.B.; Investigation, A.P.; Methodology, E.V.; Software, N.D. and D.R.; Supervision, J.-M.C., E.V. and E.B.; Writing—original draft, A.P.; Writing—review and editing, A.P., J.-M.C., E.V. and E.B. All authors have read and agreed to the published version of the manuscript.

**Funding:** This research received no external funding.

**Acknowledgments:** M. Orsel-Baldwin for GDDH13_1-1_mercator4 files processing. Pays de la Loire to funding this work (project EPICENTER).

**Conflicts of Interest:** The authors declare no competing interest.

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
