# Peer review of "Divergent DNA Methylation Signatures of Juvenile Seedlings, Grafts and Adult Apple Trees"

_2075-4655_

Round 1
Reviewer 1 Report
The manuscript from Perrin et al entitled “Divergent DNA methylation signatures of juvenile seedlings grafts and adult apple trees presents phenotypic, transcriptomic and epigenomic data comparing seedlings, grafted and adult apple trees”. The topic is very original and interesting. The plant material is of high interest and really relevant to work as well as the scientific question here.. While, the topic is really enthusiastic, reading the manuscript leads to several questions dealing with the experimental design, the statistical analyses, the arbitrary choices of thresholds and the too speculative and correlative approach to give conclusions that seem not really supported here. The manuscript redaction is also sometimes too much reductionist among hypothesis, correlation and interpretation of the literature. Sometimes, information of first interest is in sup material or difficult to find. The discussion is really slight and the conclusion was placed after the material and methods with a too speculative view of the paper and schematic figure. Overall, I have some major concerns; maybe the authors could easily answer or complete the analysis…. To improve the clarity of the real input of their work without being too much speculative.
One general question ... I can see the interest to compare grafted trees to the original adult tree but why comparing grafted to seedling? In the best case, they can compare two different biological materials (for many reasons) but what will be the interpretation of genomic data? Cause or consequence of what? I miss the idea… or it is just descriptive and then speculative? It is just one organ and one developmental stage…. Please explain and justify the precise scientific questions that can be solved with this analysis. Abstract is really not clear….the same for the introduction…. (Example: However, for the CHH sequence context is no such template exists that may allow the DNA methylation maintenance mechanism?) Just see the 2 first sentences…. They oversimplified the context and propose an opposition among annual and perennials that is uncomplete (example : From an epigenetic point of view, perennial plants are of particular interest as they have the potential to accumulate epigenetic modifications throughout their lifetime and may pass this information to the next generation… why not annuals?) Please rewrite with clear facts and objectives, results…and conclusion support by the data. Some sentences need also a confirmation by looking to the actual literature (example: However, still little is known about heritable transmission of epigenetic marks in crops and more specifically in woody perennials like apple.)… Better to give the actual support than to state about little is known. Experimental design: table S1 gives various sampling among conditions. How authors can justify comparison and statistics? With 2 repeats? Or 3 or 4? Results: Lines 124-131: where are the supporting data? It is written “We found that “? Results: Statistical analyses of DEGs and DMRs are not clear to me. One very surprising results is that authors obtain a huge number of DEGs and DMRs while they argue that globally there are not so much differences among trees and not whole genome remodeling. For DEGs, in several studies it is found much lower numbers. I expect that analyses here were really not stringent and it could help to better understand the biological message to select highly DEGs to moderate ones. For example how statistics were done compare to Breitling R, Armengaud P, Amtmann A, Herzyk P. 2004. Rank products: a simple, yet powerful, new method to detect differentially regulated genes in replicated microarray experiments. FEBS Letters 573, 83–92. The test could be run with permutations and corrected for multiple comparison errors. The DEG selection threshold was set at a false prediction rate <0.05 and a log2(fold-change) >| 2 or ?| can be used. How the differences in biological repeats were tested here? For the annotation: would it give similar results to introduce enrichment of GO (Fisher test or other?), KEGG pathway? To improve the annotation authors could use homologs in Arabidopsis and confirm the biological significance? Does microarray analysis give significant data for all genes in all conditions? Or how many are really analyzed in all conditions? Similarly for DMRs the threshold are arbitrary fixed…. But I do not really understand how and why. Looking to the results the author identify between 90 and 99.8% of CHH DMRs. It seems totally overestimated. It is well known that CHH methylation is very low (under 10%) compare to CG and CHG methylation. It seems that here the methods used are totally un-adapted to discriminate this low signal by giving too much false positive DMRs. I was surprised to read that covering was only of 3… it will be important to repeat with covering 7 to 10 and see what happens…. And use some other methodologies for CHH DMRs. According to that, I propose that we need highly expressed DEGs and their biological GO enrichment and KEGG analyses (apple and Arabidopsis)… as well as new CHH determination with better covering (7-10) and more stringent determination of CHH …. Before any analysis of the biological significance of these data. For the Scatterplot analysis, I do not see any statistical approach? … Why not testing correlations? Linear, rank, why not trying ACP with data to discriminate biological samples? Or clustering? Then, discussion could be reevaluated. The conclusion found after the materials and methods (?) is too speculative.Author Response
Comment: The manuscript from Perrin et al entitled “Divergent DNA methylation signatures of juvenile seedlings grafts and adult apple trees presents phenotypic, transcriptomic and epigenomic data comparing seedlings, grafted and adult apple trees”. The topic is very original and interesting. The plant material is of high interest and really relevant to work as well as the scientific question here.. While, the topic is really enthusiastic, reading the manuscript leads to several questions dealing with the experimental design, the statistical analyses, the arbitrary choices of thresholds and the too speculative and correlative approach to give conclusions that seem not really supported here. The manuscript redaction is also sometimes too much reductionist among hypothesis, correlation and interpretation of the literature. Sometimes, information of first interest is in sup material or difficult to find. The discussion is really slight and the conclusion was placed after the material and methods with a too speculative view of the paper and schematic figure. Overall, I have some major concerns; maybe the authors could easily answer or complete the analysis…. To improve the clarity of the real input of their work without being too much speculative.
Comment: One general question ... I can see the interest to compare grafted trees to the original adult tree but why comparing grafted to seedling? In the best case, they can compare two different biological materials (for many reasons) but what will be the interpretation of genomic data? Cause or consequence of what? I miss the idea… or it is just descriptive and then speculative? It is just one organ and one developmental stage…. Please explain and justify the precise scientific questions that can be solved with this analysis. Abstract is really not clear….the same for the introduction…. (Example: However, for the CHH sequence context is no such template exists that may allow the DNA methylation maintenance mechanism?) Just see the 2 first sentences…. They oversimplified the context and propose an opposition among annual and perennials that is uncomplete (example : From an epigenetic point of view, perennial plants are of particular interest as they have the potential to accumulate epigenetic modifications throughout their lifetime and may pass this information to the next generation… why not annuals?) Please rewrite with clear facts and objectives, results…and conclusion support by the data.
Comment: Some sentences need also a confirmation by looking to the actual literature (example: However, still little is known about heritable transmission of epigenetic marks in crops and more specifically in woody perennials like apple.)… Better to give the actual support than to state about little is known.
Authors Answer: We have improved and clarified the abstract and the manuscript in order to better explain the interest in comparing seedlings and young grafts. This specific comparison is very important as we are comparing genetically identical material which either went through grafting (asexual reproduction) and through seed formation (asexual reproduction). This allows us to better understand how DNA methylation patterns are transmitted in these two scenarios. We improved the manuscript with appropriate references lines 93-95: "However, still little is known about heritable transmission of epigenetic marks in crops and more specifically in woody perennials like apple [29, 50, 51]".
Comment: Experimental design: table S1 gives various sampling among conditions. How authors can justify comparison and statistics? With 2 repeats? Or 3 or 4?
Authors Answer: In the process of DMR calling we first merge all biological replicates of one condition. Then pairwise comparisons are performed between the conditions. So if there are 30 cytosines in a windows and that we compare one sample consisting in 2 biological replicates against second sample consisting in 3 biological replicates, statistical test implemented in the pipeline will compare the 30*2=60 cytosines of the first sample against 30*3=90 cytosines of the second sample. Perform a statistical test with different number of observations doesn’t induce a bias in the analysis. In the second step of the pipeline, the variance between replicates was calculated. A different number of biological replicates between sample doesn’t induce any bias here, too. But higher number of biological replicates in a sample give more power and confidence to variance. So, the number of replicates do not have an impact on the data analysis.
Comment: Results: Lines 124-131: where are the supporting data? It is written “We found that “?
Authors Answer: We modified the manuscript as following (line 126-127): “Unlike most heterozygous apple varieties, the GDDH13 doubled haploid apple shows a high self-compatibility level”.
Comment :Results: Statistical analyses of DEGs and DMRs are not clear to me.
One very surprising results is that authors obtain a huge number of DEGs and DMRs while they argue that globally there are not so much differences among trees and not whole genome remodeling.
Authors answer: We are aware that this number is high, compared to DMRs in the CG and CHG contexts. In order to clarify our findings: we found 98782 DMRs in the CHH context for SDvsOG and 42694 for YGvsOG but only 179 in the CHG context in SDvsOG, and 389 in YGvsOG and 479 in the CG context in SDvsOG and 918 in YGvsOG. The DMRs we identified consist of 200pb sliding window regions with 100pb overlap. That means that in the apple genome (643.2MB) there are approximatively 6.5 million windows that have been tested. Overall, in the SDvsOG comparison, we identified DMRs representing 1.5% of all possible 200bp windows, while in the YGvsOG comparison, we found DMRs representing 0,66% of all possible 200bp windows. This low percentage of 200bp regions in which a DMR was identified is the reason why we conclude that there was no global methylation change on the genome overall.
Comment :For DEGs, in several studies it is found much lower numbers. I expect that analyses here were really not stringent and it could help to better understand the biological message to select highly DEGs to moderate ones.
Authors Answer: Microarray analyses were performed as in Celton et al 2014 (ref in the bottom of this document). Data were first normalized with the lowess method. Normalized intensity values were then subtracted from the background to provide an estimation of the transcript expression levels. Differential expression analyses were performed using the lmFit function and the Bayes-moderated Student’s t test using the package LIMMA (Smyth, 2005) from the Bioconductor project. Genes were considered significantly differentially transcribed if their P values were P , 0.01 for the dye-switch experiment. Genes differentially transcribed were then screened using a threshold fold change of one or greater.
Comment: For example how statistics were done compare to Breitling R, Armengaud P, Amtmann A, Herzyk P. 2004. Rank products: a simple, yet powerful, new method to detect differentially regulated genes in replicated microarray experiments. FEBS Letters 573, 83–92. The test could be run with permutations and corrected for multiple comparison errors. The DEG selection threshold was set at a false prediction rate <0.05 and a log2(fold-change) >| 2 or ?| can be used.
Authors Answer: in this analysis we used a p-value<0.01. Within this list, all differentially transcribed genes had an FDR < 0.1
Comment: How the differences in biological repeats were tested here?
Authors Answer: As mentioned in Methods, “Analyses were performed using the R language (R Development Core Team, 2009); data were normalized with the lowess method, and differential transcription analyses were performed using the lmFit function and the Bayes moderated test using the package LIMMA [89]. “
Comment: For the annotation: would it give similar results to introduce enrichment of GO (Fisher test or other?), KEGG pathway? To improve the annotation authors could use homologs in Arabidopsis and confirm the biological significance?
Authors Answer: Arabidopsis annotation was indeed used to annotate the apple genome. Because of the duplicated nature of the apple genome, we then used the Arabidopsis homologues for further analyses in Mapman. Mapman was also used to perform enrichment test on the DEG. As mention in Material and Methods we use the BH correction implemented in Mapman as statistical test.
Comment: Does microarray analysis give significant data for all genes in all conditions? Or how many are really analyzed in all conditions?
Authors Answer: The microarray design contains all protein-coding and non-protein coding genes annotated in apple, and a representative portion of transposable elements. The apple genome gene annotation was performed using RNA sequencing data generated from over 15 different tissues/organs and development stages. This was further complemented by an automatic annotation pipeline trained on the transcripts identified via RNA sequencing. As such, we believe that the microarray design is representative of the apple transcriptome.
Comment: Looking to the results the author identify between 90 and 99.8% of CHH DMRs. It seems totally overestimated. It is well known that CHH methylation is very low (under 10%) compare to CG and CHG methylation. It seems that here the methods used are totally un-adapted to discriminate this low signal by giving too much false positive DMRs.
Authors Answer:: Indeed, between 90 and 99.8% of the identified DMRs were in the CHH context.
For the SDvsOG, we found 99437 DMRs. Each DMR is calculated on the basis of a 200bp window. In total, there are 6.432 million possible windows (each window overlaps with its neighbor by 100bp). Hence, for the SDvsOG, we identified DMRs for 1.54% of all the possible windows. Within these 99437 DMRs, 99.3% were found in the CHH context, thus representing only 1.53% of all possible windows.
Furthermore, since the CHH context is generally found to have a low methylation level, a small increase or decrease (when confirmed on all biological replicates) will quickly be identified as significantly different. For example, if on a 200 pb windows the methylation in CHH context is about 10% in OG and 6% in SD and the methylation in CG context is about 50% in OG and 46% in SD the difference between OG and SD is 4% in both context, but the 4% represent a relatively higher methylation variation in CHH context (40% decrease) than in CG (8% decrease).
Comment: I was surprised to read that covering was only of 3… it will be important to repeat with covering 7 to 10 and see what happens…. And use some other methodologies for CHH DMRs. According to that, I propose that we need highly expressed DEGs and their biological GO enrichment and KEGG analyses (apple and Arabidopsis)… as well as new CHH determination with better covering (7-10) and more stringent determination of CHH …. Before any analysis of the biological significance of these data. + Similarly for DMRs the threshold are arbitrary fixed…. But I do not really understand how and why.
Authors answer: In order to clarify this point, we improved the Material and Methods section (lines 530-552). WGBS were performed with the aim to obtain 13X coverage per sample. The coverage of 3 was set as the minimum coverage in order to perform the DMRs calculation,
Moreover, we used SDA (standard deviation of averages) to filter our data, this allowed us to remove all DMRs were 1 sample was not consistent with others. And we also applied a stringent threshold of 1% p-value to reduce the rate of detection of false positives.
Threshold to further filtered DMRs where fixed by extensive preview DMRs on JBrowse. We fixed SDA threshold when we can’t detect any more differences in methylation level between sample to a calculated DMRs.
Comment: For the Scatterplot analysis, I do not see any statistical approach? … Why not testing correlations? Linear, rank, why not trying ACP with data to discriminate biological samples? Or clustering?
Authors answer: We have tested correlations as recommended. We used linear test and performed a Student test. As now displayed in figure 7, we only found a correlation for DEG-DMRs in gene promoters and terminators in the SDvsOG comparison. We corrected the manuscript according to this finding (figure 7 and line 372-375): “Our data also indicate that, independently of the hypomethylated DMRs located in promoting region and in terminator region of a position relative to a gene, hypermethylated DMRs were associated with differential transcriptional regulation in the SD compared to the OG sample (Fig. 7A and B). but we didn’t find a correlation to OGvsYG comparison”.
Comment: Then, discussion could be reevaluated. The conclusion found after the materials and methods (?) is too speculative.
Authors answer: the position of the conclusion was selected following other MDPI published manuscript layout. We propose to modify the position of the conclusion to put it after discussion if the editor agrees. We also modified the too speculative part in the conclusion as recommended. We suppress “presumably allowing the seedling to increase its competitiveness” (line 485)
Reference
- Celton JM, Dheilly E, Guillou MC, et al (2014) Additional amphivasal bundles in pedicel pith exacerbate central fruit dominance and induce self-thinning of lateral fruitlets in apple. Plant Physiol 164:1930–1951. https://doi.org/10.1104/pp.114.236117
Reviewer 2 Report
Dear authors
This is a very interesting manuscript but very wordy and hard to be read by common scientific readers.
The results are comprehensible but the analysis, in my opinion, make them confusing.
If, in fact, the objective is to compare “the transmission of epigenetic marks and their potential effects on transcription during sexual and asexual reproduction in apple“ one of the samples should be the control, the base line that stays immobile with the other values moving around it.
For example: if the transcription level in Young Grafted (YG)and Seedlings (SD) is higher that Old Grafted (OG), and the last one is taken as control then YG and SD will show a higher expression level (of whatever) but the level of GD should not be referred as “lower” because this is obvious and tautological.
Another (in my opinion worse) example: Transcription of photosynthesis related genes” (please correct in the last figure) YG and SD are respectively Intermediate and Upregulated in comparison to OG. But OG should not be assumed as downregulated regarding YF and SD.
Of course OG can be assumed downregulated buy then SD and YG are not “upregulated”.
Such kind of reasoning could be used only if all three (OG, YG and SD) were exposed to any additional factor and then one could be upregulated, another downregulated and a third showing an intermediate response to the new circumstances.
In my opinion the manuscript should be revised making it more concise and clearer,
Examples of small issues
Line 57 “ However, for the CHH sequence context is (??) no such template exists that may allow the DNA methylation maintenance mechanism.”
Line 79 ” pollen sperm cell there is a decrease in RdDM activity” – Please check the cited article. After a second mitosis the pollen grains will contain TWO sperm cells.
Line 231 ”some regions displaying a higher enrichment (Fig. 4C, red boxes)”. Perhaps “ higher enrichment (Fig, 4E, red boxes)”
Lines 297/298 “We excluded DMRs associated with the CG and CHG context here analysis due of their very limited number” (does not make sense!)
Please check also for some unusual concepts., e.g. “tree multiplication”, professional tree producers would say; “tree propagation”.
Author Response
Dear authors
This is a very interesting manuscript but very wordy and hard to be read by common scientific readers.
The results are comprehensible but the analysis, in my opinion, make them confusing.
Comment: If, in fact, the objective is to compare “the transmission of epigenetic marks and their potential effects on transcription during sexual and asexual reproduction in apple“ one of the samples should be the control, the base line that stays immobile with the other values moving around it. For example: if the transcription level in Young Grafted (YG)and Seedlings (SD) is higher that Old Grafted (OG), and the last one is taken as control then YG and SD will show a higher expression level (of whatever) but the level of GD should not be referred as “lower” because this is obvious and tautological. Another (in my opinion worse) example: Transcription of photosynthesis related genes” (please correct in the last figure) YG and SD are respectively Intermediate and Upregulated in comparison to OG. But OG should not be assumed as downregulated regarding YF and SD. Of course OG can be assumed downregulated buy then SD and YG are not “upregulated”. Such kind of reasoning could be used only if all three (OG, YG and SD) were exposed to any additional factor and then one could be upregulated, another downregulated and a third showing an intermediate response to the new circumstances.
Authors answer: We understand that this a bit confusing. Following your comments we fixed the Old Graft tree as the reference and modified all the figures and text to describe SD and YG in comparison to OG.
Comment : Transcription of photosynthesis genes related è “Transcription of photosynthesis related genes” (please correct in the last figure)
Authors answer: thank you for highlighting this mistake, we corrected it.
In my opinion the manuscript should be revised making it more concise and clearer,
Examples of small issues
Comment : Line 57 “ However, for the CHH sequence context is (??) no such template exists that may allow the DNA methylation maintenance mechanism.”
Authors answer: we modified the sentence to make it clearer: “However, for the CHH sequence context, no such template exists that may allow the DNA methylation maintenance mechanism”
Comment : Line 79 ” pollen sperm cell there is a decrease in RdDM activity” – Please check the cited article. After a second mitosis the pollen grains will contain TWO sperm cells.
Authors answer: we completed the sentence “, in the central cell of the mature female gametophyte and in the matures pollens sperms cells there is a decrease in RdDM activity”
Comment : Line 231 ”some regions displaying a higher enrichment (Fig. 4C, red boxes)”. Perhaps “ higher enrichment (Fig, 4E, red boxes)”
Authors answer: in a previous version the three histograms on the right were under same letter “B”, we divided them in three but we missed this change in manuscript. Thank you for highlight it
Comment : Lines 297/298 “We excluded DMRs associated with the CG and CHG context here analysis due of their very limited number” (does not make sense!).
Authors answer: we completed the sentences: “We excluded DMRs associated with the CG and CHG contexts from further analysis due of their very limited number”
Comment : Please check also for some unusual concepts., e.g. “tree multiplication”, professional tree producers would say; “tree propagation”.
Authors answer: we modified tree multiplication in tree propagation as recommended
Reviewer 3 Report
As perennial plants may change DNA methylation and transcription over time, the manuscript investigates the influence of grafting on DNA methylation change in apple. For this, the authors use material with the double haploid GDDH13 genome and compare tissues of seedlings, young and old grafts. They find that transcription and DNA methylation in young grafts represent an intermediate stage between seedlings and the old grafts.
I think the study is timely and has value. The introduction introduces all material needed to understand the paper. The discussion is relatively clear and the conclusion helpful, especially with explanatory Figure 8. Nevertheless, I still have some issues, especially with the presentation of the results, leading me to advise a major revision.
Remarks:
1) The manuscript repeatedly refers to histograms, but shows stacked bar charts in all of these cases (Figures 4, 5, and 6).
2) Figure 2 is partially unclear: Although 720 common genes are up- and 1365 common genes are down-regulated, the heatmap ratio (Figure 2C) does not reflect these numbers. The up-regulated area is much larger than the down-regulated area, which is in conflict with the actual data. The same is true for the TEs (Figure 2D). In addition, the scalebar is unintuitive, with the negative fold change to the right and the positive to the left. I recommend switching this.
3) I don’t understand Figure 3. I can guess what is meant for Figure 3A, but for Figure 3B I cannot even guess: The Y axis goes well beyond 100 % and I just don’t understand how 140 % of all detected differentially expressed TEs can be retrotransposons (class I), and 120 % can be DNA transposons (class II). This figure has to be either better explained, better labeled or/and has to be redrawn. (This also corresponds to Figure 6).
4) Regarding the TE annotation, it would be helpful, if the categories were described in more detail. What exactly is “other” than class I and class II?
5) The text references to Figure 4 do not match the actual Figure (e.g. Figure 4C is referenced, but Figure 4E is described.)
6) Figure 4E/ Figure S2:
a) How have the DMR hot spots been evaluated?
b) The CG circle is violet and not red.
7) In Table 1, the indicated percentages refer to different reference systems (e.g. all cytosines vs. individual cytosine contexts). This renders the table very confusing. Maybe the authors want to remove some of the percentages.
8) Figure 5 and the corresponding manuscript text:
a) Line 250 to 280 is relatively cryptic. It would be highly recommended to translate this to more plain English.
b) As a result, Figure 5 did also not become clear. It would be very helpful to explain the Y axis (delta mC – OG [%]). What exactly was subtracted and why? Some of the values do not make any sense to me, e.g. in the CHG context (OGvsSD): Gene-DMRs=13, TE-DMRs=38, All-DMRs=22. As I understand it, the All-DMR values should add up or at least should be bigger than the sum of the other two values. I recommend thinking about an intuitive representation or clearer explanation of the data.
9) In many instances, but not all, the manuscript uses a “comma” instead of the “decimal point”. This should be corrected.
10) The manuscript contains many grammar mistakes and typos. It should be carefully proofread again. I noted some examples, but I cannot include all in this report:
a) The material and methods section is in most cases clear and contains all information to reproduce the experiments. However, this section contains many grammar errors, obfuscating the content in some instances, e.g. at line 523 to 525, the meaning is completely unclear.
b) Line 13: double word “rapidly”
c) Line 57: “is” has to be removed
d) Line 58: “guide the [..] machinery TO regions”
e) Line 118: “independent”, not “independently”
f) Line 153: “genes”, not “gene”…
(and many more, especially in the M&M section).
Author Response
As perennial plants may change DNA methylation and transcription over time, the manuscript investigates the influence of grafting on DNA methylation change in apple. For this, the authors use material with the double haploid GDDH13 genome and compare tissues of seedlings, young and old grafts. They find that transcription and DNA methylation in young grafts represent an intermediate stage between seedlings and the old grafts.
I think the study is timely and has value. The introduction introduces all material needed to understand the paper. The discussion is relatively clear and the conclusion helpful, especially with explanatory Figure 8. Nevertheless, I still have some issues, especially with the presentation of the results, leading me to advise a major revision.
Remarks:
1) The manuscript repeatedly refers to histograms, but shows stacked bar charts in all of these cases (Figures 4, 5, and 6).
Authors answer: Indeed we misused the work “histogram” in our figures. We modified the figures caption of our manuscript according to your advice.
“Figure 4: Global overview of DNA methylation differences between SD, YG and OG. (A) clustered column chart presenting the genome wide cytosine methylation level (in percentage) of the three methylations context (CG, CHG and CHH). Student test was performed to evaluate differences but none was found to be significant. B, C and D. stacked graph representing the number of DMRs for each comparison: hypomethylated (above 0, in blue) or hypermethylated (below 0, in orange) in the SD and YG samples compare to OG for all DMRs (B), Gene-DMRs (C) and TE-DMRs (D). DMRs in all sequence contexts were counted and values are indicated in graph. (E) density plot of number of DMRs in 50 kb windows on the GDDH13 genome for OGvsSD (see supplemental figure S1 for OGvsYG). In violet, DMRs in the CG context, in blue for the CHG context and in orange the CHH context. Each point represents the number of DMRs in a 50kb window of the genome. Red dashed boxes indicate the presence of DMR hot spots.
Figure 5: Levels of DNA methylation changes in gene and TE annotations. Stacked graph depicting DMR methylation variations (δmC) between samples separated by sequence context and functional annotation. All DMRs are presented in the All-DMRs column, genes and TEs DMRs in the Gene-DMRs and TE-DMRs, respectively. DMRs were filtered by p-value and SDA (standard deviation average) in accordance to a fixed threshold (Table S2). Student test was performed to evaluate differences in δmC, results are represented by an asterix depending on the p-value threshold: *: 5%; **: 1%; ***: 1‰. δmC: delta of methylation. In blue and orange, the hypomethylated and hypermethylated DMRs in SD and YG compare to OG respectively.
Figure 6: Classification of differentially transcribed genes that are associated to DMRs. Stacked graph depicting the percentage of DTG-DMRs (A and B) and DTTE-DMRs (C and D) in the respective comparisons in function of gene or TEs classification. Only DMRs in the CHH context are presented here. In (A) and (C) all DTGs- and DTTEs-DMRs were used while in (B) and (D) we only considered DTGs and DTTEs with differential transcription ratio greater than 1.5 in absolute value. Gene classes representing less than 5% (A) or 10% (B) of the total in the three conditions were summed up in “other class”.”
2) Figure 2 is partially unclear: Although 720 common genes are up- and 1365 common genes are down-regulated, the heatmap ratio (Figure 2C) does not reflect these numbers. The up-regulated area is much larger than the down-regulated area, which is in conflict with the actual data. The same is true for the TEs (Figure 2D). In addition, the scalebar is unintuitive, with the negative fold change to the right and the positive to the left. I recommend switching this.
Authors answer: We inverted the blue and the red color between A and B to C and D. We corrected it and also switched the scale as recommended.
3) I don’t understand Figure 3. I can guess what is meant for Figure 3A, but for Figure 3B I cannot even guess: The Y axis goes well beyond 100 % and I just don’t understand how 140 % of all detected differentially expressed TEs can be retrotransposons (class I), and 120 % can be DNA transposons (class II). This figure has to be either better explained, better labeled or/and has to be redrawn. (This also corresponds to Figure 6).
Authors answer: Following your comment we modified the stacked graph into a clustered column graph that is clearer.
4) Regarding the TE annotation, it would be helpful, if the categories were described in more detail. What exactly is “other” than class I and class II?
Authors answer: The “other” category was used following Daccord et al (2017) TE classification, We add the following sentence in Figure 3 description:” Transposon in “other” class correspond to, potential host gene and unclassified TEs according to Daccord et al. (2017).” .
5) The text references to Figure 4 do not match the actual Figure (e.g. Figure 4C is referenced, but Figure 4E is described.)
Authors answer: Indeed, in a previous version the three histograms on the right were under same letter “B”. We have made the appropriate changes. Thank you for highlighting it.
6) Figure 4E/ Figure S2:
- a) How have the DMR hot spots been evaluated?
- b) The CG circle is violet and not red.
Authors answer:
- a) We considered DMR hotspots as regions containing at least three DMRs in at least two sequence contexts in a 15 kb window.
- b) We modified the color legend according to your comment
7) In Table 1, the indicated percentages refer to different reference systems (e.g. all cytosines vs. individual cytosine contexts). This renders the table very confusing. Maybe the authors want to remove some of the percentages.
Authors answer: We modified the table by addition of color to percentage and add following sentences in table 1 description: “Percentage in grey color correspond to the percentage of DMRs in specific cytosine context in the comparison. Percentage in blue color correspond to the percentage of all DMRs in the comparison.”
8) Figure 5 and the corresponding manuscript text:
- a) Line 250 to 280 is relatively cryptic. It would be highly recommended to translate this to more plain English.
- b) As a result, Figure 5 did also not become clear. It would be very helpful to explain the Y axis (delta mC – OG [%]). What exactly was subtracted and why? Some of the values do not make any sense to me, e.g. in the CHG context (OGvsSD): Gene-DMRs=13, TE-DMRs=38, All-DMRs=22. As I understand it, the All-DMR values should add up or at least should be bigger than the sum of the other two values. I recommend thinking about an intuitive representation or clearer explanation of the data.
Authors answer :
- a) We make some modification in this part.
- b) the indicated number represented the numbers of gene-DMRs, TE-DMRs or all-DMRs considered to create the plot. Numbers of DMRs in “All-DMRs” is greater that sum of DMRs in “gene-DMRs” and “TE-DMRs” because there are some DMRs located in other region than in gene or TE context.
9) In many instances, but not all, the manuscript uses a “comma” instead of the “decimal point”. This should be corrected.
Authors answer : thank you to highlight this problem, we now fixed all this issues
10) The manuscript contains many grammar mistakes and typos. It should be carefully proofread again. I noted some examples, but I cannot include all in this report:
- a) The material and methods section is in most cases clear and contains all information to reproduce the experiments. However, this section contains many grammar errors, obfuscating the content in some instances, e.g. at line 523 to 525, the meaning is completely unclear.
- b) Line 13: double word “rapidly”
- c) Line 57: “is” has to be removed: However, for the CHH sequence context no such template exists that may allow the DNA methylation maintenance mechanism
- d) Line 58: “guide the [..] machinery TO regions”: to guide the DNA methylation machinery to regions with sequence homology to the siRNAs.
- e) Line 118: “independent”, not “independently”: We present evidence that globally, genome-wide DNA methylation levels are stable in apple independent of the mode of multiplication
- f) Line 153: “genes”, not “gene”… : In total these DEGs DTGs include 13,.5% of all annotated genes on the microarray for OGvsSD
(and many more, especially in the M&M section).
Authors answer : We again carefully read the methods in order to corrected maximum of issues